# Reproductive, Obstetric and Neonatal Outcomes in Women with Congenital Uterine Anomalies: A Systematic Review and Meta-Analysis

**DOI:** 10.3390/jcm10214797

**Published:** 2021-10-20

**Authors:** Min-A Kim, Hyo Sun Kim, Young-Han Kim

**Affiliations:** 1Department of Obstetrics and Gynecology, Gangnam Severance Hospital, Institute of Women’s Life Medical Science, Yonsei University College of Medicine, Seoul 03722, Korea; makim302@yuhs.ac (M.-A.K.); KHS88@yuhs.ac (H.S.K.); 2Department of Obstetrics and Gynecology, Severance Hospital, Institute of Women’s Life Medical Science, Yonsei University College of Medicine, Seoul 06273, Korea

**Keywords:** congenital uterine anomalies, fertility, miscarriage, obstetric outcome, neonatal outcome

## Abstract

Congenital uterine anomalies (CUA) may influence reproductive performance, resulting in adverse pregnancy associated complications. This study aimed to assess the association of CUA subtypes with reproductive, obstetric, and perinatal outcomes. We performed a systematic search of the MEDLINE, EMBASE, and Cochrane libraries for studies comparing pregnancy outcomes between women with CUA and those with a normal uterus. The random effects model was used to estimate the odds ratios (ORs) with a 95% confidence interval (CI). Women with CUA had a lower rate of live births (OR 0.47; 95% CI 0.33–0.69), and a higher rate of first trimester miscarriage (OR, 1.79; 95% CI 1.34–2.4), second trimester miscarriage (OR 2.92; 95% CI 1.35–6.32), preterm birth (OR 2.98; 95% CI 2.43–3.65), malpresentation (OR 9.1; 95% CI 5.88–14.08), cesarean section (OR 2.87; 95% CI 1.56–5.26), and placental abruption (OR 3.12; 95% CI 1.58–6.18). Women with canalization defects appear to have the poorest reproductive performance during early pregnancy. However, unification defects were associated with obstetric and neonatal outcomes throughout the course of pregnancy. It may be beneficial for clinicians to advise on potential complications that may be increased depending on the type and severity of CUA.

## 1. Introduction

The prevalence of congenital uterine anomalies (CUA) varies significantly, with reports ranging from 0.06 to 38% [1,2,3,4,5,6,7,8]. It can be inferred that these wide variations between studies are due to the inaccuracy of diagnostic tests, the use of different diagnostic techniques, differences in the evaluated patient population, and nonstandardized classification systems. Moreover, in many cases of CUA, it is difficult to detect because of a lack of awareness; therefore, the actual distribution and frequency of CUA occurrence in the general population are not accurately known. As medical attention starts with dysfunction (such as miscarriage or infertility), most studies have reported an increased prevalence of CUA in patients with reproductive problems.

The female reproductive tract differentiates from two Müllerian ducts, which develop within the first six weeks of fetal life [9,10]. Normal development of the female reproductive tract occurs through multistep processes, such as differentiation, migration, fusion, and subsequent absorption of the Müllerian system [11]. CUA are caused by an abnormal interruption during this development, and may result in an inability to conceive. The absence of a universally accepted classification system for CUA is problematic because such a system would allow physicians to categorize the symptomatology, treatment, and outcome in affected patients. The American Fertility Society (AFS) classification (1988) is the most commonly used classification, over the past three decades, for the categorization of CUA and is also used in most of the studies. Arcuate uterus is the mildest form of resorption failure or normal variation [12,13,14]. In arcuate uterus, the uterine cavity displays a concave contour toward the fundus and is not considered clinically relevant. Depending on the failure of organogenesis and varying degrees of fusion or absorption defects, CUA can be divided into unification defects of the Müllerian ducts (unicornuate, bicornuate, or didelphys uterus) and canalization defects from incomplete resorption of the midline septum (subseptate or septate uterus) [15].

The endometrial cavity is a space for successful embryo implantation and placenta formation, and defects in endometrial cavity formation may lead to infertility, recurrent miscarriage, and adverse pregnancy outcomes [16,17]. CUA vary, and not all patients have clinical symptoms. The pregnancy rate of women with CUA is not much different from that of women with a normal uterus, and pregnancy can be well maintained and lead to normal delivery. However, the frequency of obstetrical complications, such as miscarriage, preterm birth, intrauterine growth restriction (IUGR), and malpresentation, is high, depending on the type and severity of CUA [18,19]. Moreover, symptoms in CUA patients may include nonperiodic pelvic pain, menstrual pain, abnormal vaginal bleeding, and extrauterine pregnancies [9]. 

Our study aimed to evaluate the association of CUA with reproductive, obstetric, and neonatal outcomes in women, and determine whether CUA subtypes have a specific impact and the extent of the impact on a wide range of reproductive outcomes. To this end, the existing literature was thoroughly reviewed, and an updated meta-analysis was performed.

## 2. Materials and Methods

### 2.1. Search Strategy

We performed a literature search for all articles published until May 2021, and written in English, that reported comparisons of reproductive outcomes between women with CUA and women with a normal uterus. Two reviewers (Kim MA and Kim YH) independently performed an online systematic search using the MEDLINE, EMBASE, and Cochrane libraries. For the search, we combined the Medical Subject Headings (MeSH) and text terms covering CUA and reproductive outcomes. Appendix A outlines the full search strategies for each database. The reviewers manually searched and cross referenced the review articles found by electronic searches to locate additional cited articles that were missing from our online searches. 

### 2.2. Selection Criteria

The inclusion criteria for this study were as follows: studies with an appropriate control group, and studies comparing reproductive outcomes between women with a diagnosis of CUA and women with a normal uterus. For the pregnancy outcomes of interest, the rate of clinical pregnancy (defined as the presence of an intrauterine gestational sac on sonography) among the total number of women, first trimester miscarriage, second trimester miscarriage, and recurrent pregnancy loss were investigated as reproductive outcomes. Live births, ectopic pregnancy, preterm birth, preterm premature rupture of membranes (PPROM), malpresentation, cesarean section, preeclampsia, placental abruption, postpartum hemorrhage, cervical incompetence, intrauterine fetal death (IUFD), placenta previa were investigated as obstetric outcomes. IUGR or small for gestational age (SGA; birth weight below the 10th percentile for the gestational age), low birth weight (LBW; <2500 g), and perinatal mortality among the total number of clinical pregnancies were investigated as neonatal outcomes. If the definition of the outcome was not specified, the classification of authors was based on a generalized definition. Studies that met the following criteria were excluded from the analysis: case reports and series, review articles, studies with no or an inappropriate control group, studies where data extraction was impossible, and studies reporting pregnancy outcomes in the same women before and after metroplasty. We performed data extraction for each subtype of CUA. 

### 2.3. Study Selection

Two independent authors screened the titles and abstracts, and full articles were selected if the study met the selection criteria and reported the reproductive outcomes between women with CUA and those with a normal uterus. Full articles were retrieved for clarity, unless explicitly stated in the abstract. Studies unavailable online were collected directly from institutional libraries and references were cross checked to find additional articles. If more than one study by the same group of investigators describing the same population was published, the most recent version or the most comprehensive publication was used in the final meta-analysis.

### 2.4. Data Extraction

We extracted the following information from each article: name of authors, journal, study design, year of publication, country of patient, selection criteria, sample size of each group, mode of conception, type of CUA, diagnostic technique, classification of CUA, and data analyzed for reproductive outcomes. 

### 2.5. Quality Assessment

We assessed the quality of the included studies using the Newcastle–Ottawa Scale (NOS). NOS evaluates three quality parameters: selection of the study groups, comparability of groups, and ascertainment of either the exposure or outcome of interest for case-control or cohort studies, respectively. Each study had a maximum of nine points, giving scores for various questions in each category, and those with a score of 6 or more were considered high quality studies.

### 2.6. Statistical Analysis

This systematic review and meta-analysis followed the Preferred Reporting Items for Systematic Reviews and Meta-Analyses (PRISMA) guidelines [20] and the guidelines provided by the Cochrane Handbook for Systematic Reviews [21]. This study was not registered with the PROSPERO database. Reproductive outcomes were reported as odds ratios (ORs), representing the odds of adverse reproductive outcomes for women with CUA compared to women with a normal uterus. We used the random effects model to calculate pooled ORs and 95% confidence intervals (CI) [22]. We performed a subanalysis to analyze the differences in outcomes by the type of CUA. We considered I^2^ values > 25%, >50%, and >75% as evidence of low, moderate, and severe statistical heterogeneity, respectively. Sensitivity analysis was conducted to evaluate studies with a dominant effect on the meta-analysis or by excluding one study each time and assessing the effect on the main summary estimate. Statistical significance was set at *p* < 0.05. We performed all statistical analyses using Review Manager (RevMan) 5.4 (The Nordic Cochrane Centre: Copenhagen, Denmark).

## 3. Results

### 3.1. Study Characteristics

The systematic search retrieved 6888 citations in total, of which 1035 were excluded as duplicates. A total of 256 studies were considered potentially eligible for full text review by reviewing the titles and abstracts of the remaining 5853 manuscripts. Of these 256 studies, following the scrutiny of each manuscript, we identified 37 relevant studies [23,24,25,26,27,28,29,30,31,32,33,34,35,36,37,38,39,40,41,42,43,44,45,46,47,48,49,50,51,52,53,54,55,56,57,58,59] that met the inclusion criteria as eligible for the meta-analysis (Figure 1). 

The baseline characteristics of all the included studies are shown in Table 1. Of the 37 studies included, only 5 were prospective studies, whereas the remaining 32 were retrospective studies. From these 37 studies, a total of 7053 women were identified as having CUA and 701,527 women with a normal uterus. All studies included women with different types of CUA. Of these 37 studies, 30 reported results for each specific subtype of CUA, whereas seven reported the results of the entire CUA without distinction of a particular subtype. The diagnostic methods for CUA also varied between the studies. They included hysterosalpingography (HSG), 2 dimensional (2D) or 3 dimensional ultrasound (3D-US), hysteroscopy, sonohysterography, laparoscopy, laparotomy, during cesarean section, and MRI. Each diagnostic method was used alone or in combination to confirm the CUA. The classification system used in each study to define the subtypes of CUA was inconsistent, but approximately half of the included studies used the classification of the AFS. Concerning the mode of conception, 13 studies included women who underwent assisted reproductive technologies (ART), two studies with natural conceptions (NC), seven studies with ART/NC, and 15 studies did not mention this. The median NOS for 37 studies was 7 (range, 6–9), and all included studies were considered high quality (Appendix A).

### 3.2. Reproductive Outcomes

#### 3.2.1. Clinical Pregnancy Rate

Sixteen studies compared the clinical pregnancy rate in 13,478 women, consisting of 2659 women with CUA and 10,819 women with a normal uterus. When all CUA subtypes were combined, there were no significant differences in the clinical pregnancy rate between women with CUA and the control group (OR, 0.87; 95% CI 0.7–1.08; *p* = 0.2; I^2^ = 70%; Figure 2A). In the subgroup analysis by CUA subtype (Table 2), pooled results in women with an arcuate uterus showed no significant difference in clinical pregnancy rate compared to that in women with a normal uterus (OR, 1.0; 95% CI 0.64–1.58; *p* = 0.99; I^2^ = 73%; 6 studies). In the canalization defects (OR 0.59; 95% CI 0.32–1.08; *p* = 0.09; I^2^ = 52%; seven studies), the clinical pregnancy rate decreased significantly in septate uterus (OR 0.45; 95% CI 0.21–0.95; *p* = 0.04; I^2^ = 31%; six studies) but not in subseptate (OR 0.73; 95% CI 0.28–1.92; *p* = 0.53; I^2^ = 60%; five studies). In the unification defects (OR 0.72; 95% CI 0.57–0.9; *p* = 0.005; I^2^ = 40%; 10 studies), there were no significant differences in clinical pregnancy rate between women with bicornuate uterus (OR 0.57; 95% CI 0.32–1.03; *p* = 0.06; I^2^ = 3%; five studies) or didelphys uterus (OR 0.36; 95% CI 0.09–1.39; *p* = 0.14; I^2^ = 0%; three studies) than that in women with a normal uterus. However, a meta-analysis of women with unicornuate uterus showed a significant decrease in the clinical pregnancy rate (OR 0.75; 95% CI 0.58–0.99; *p* = 0.04; I^2^ = 59%; nine studies).

#### 3.2.2. Live Birth

Fifteen studies compared the rate of live births in a total of 11,113 pregnancies, comprising 1538 pregnancies in women with CUA and 9575 pregnancies in women with a normal uterus. The pooled analysis showed a lower rate of live birth in women with CUA (OR, 0.47; 95% CI, 0.33–0.69; *p* < 0.001; I^2^ = 82%; Figure 2B) than that in women with a normal uterus. In subgroup analysis according to the subtype of CUA (Table 2), there was a significant decrease in live birth rate in women with arcuate uterus (OR, 0.45; 95% CI, 0.22–0.92; *p* = 0.03; I^2^ = 88%; five studies), septate uterus (OR 0.25; 95% CI 0.09–0.75; *p* = 0.01; I^2^ = 49%; four studies), and unicornuate uterus (OR 0.57; 95% CI 0.34–0.96; *p* = 0.04; I^2^ = 69%; 10 studies), but not in other types of CUA.

#### 3.2.3. First Trimester Miscarriage

Thirteen studies reported data on first trimester miscarriage in a total of 81,024 pregnancies, which comprised 2645 pregnancies in women with CUA and 78,379 pregnancies in women with a normal uterus. The pooled analysis showed that all women with CUA had a higher rate of first trimester miscarriage than that in women with a normal uterus (OR 1.79; 95% CI 1.34–2.4; *p* <0.001; I^2^ = 70%; Figure 2C). Subgroup analysis according to the subtype of CUA (Table 2) showed that the risk of first trimester miscarriage was not significantly different between women with an arcuate uterus and those with a normal uterus (OR 1.38; 95% CI 0.88–2.17; *p* = 0.16; I^2^ = 45%; six studies). A meta-analysis of canalization defects (OR 3.32; 95% CI 1.96–5.6; *p* < 0.001; I^2^ = 64%; six studies) revealed that subseptate uterus (OR 4.36; 95% CI 2.64–7.21; *p* < 0.001; I^2^ = 15%; four studies) and septate uterus (OR 2.55; 95% CI 1.33–4.91; *p* = 0.005; I^2^ = 58%; five studies) had an increased risk of first trimester miscarriage. The meta-analysis of unification defects (OR 1.77; 95% CI 1.18–2.65; *p* = 0.006; I^2^ = 54%; eight studies) revealed that only in bicornuate uterus (OR 2.59; 95% CI 1.25–5.35; *p* = 0.01; I^2^ = 65%; six studies) had a higher risk of first trimester miscarriage, but not in unicornuate uterus (OR, 1.45; 95% CI, 0.85–2.48; *p* = 0.17; I^2^ = 37%; seven studies) and didelphys uterus (OR 1.26; 95% CI 0.33–4.76; *p* = 0.73; I^2^ = 67%; four studies). 

#### 3.2.4. Second-Trimester Miscarriage

Eleven studies reported data on second trimester miscarriage in 80,347 pregnancies, consisting of 2592 pregnancies in women with CUA and 77,755 pregnancies in women with a normal uterus. The meta-analysis indicated that all women with CUA had a higher rate of second trimester miscarriage than that in women with a normal uterus (OR 2.92; 95% CI 1.35–6.32; *p* = 0.006; I^2^ = 87%; Figure 2D). The subgroup analysis by type of CUA (Table 2) showed that women with an arcuate uterus (OR 2.01; 95% CI 1.03–3.93; *p* = 0.04; I^2^ = 11%; five studies) had a higher rate of second trimester miscarriage than that in women with a normal uterus. In canalization defects (OR 3.38; 95% CI 1.94–5.88; *p* < 0.001; I^2^ = 18%; five studies), second trimester miscarriage increased significantly in septate uterus (OR 4.33; 95% CI 2.52–7.43; *p* < 0.001; I^2^ = 0%; four studies), but not in subseptate uterus (OR 1.9; 95% CI 0.54–6.75; *p* = 0.32; I^2^ = 41%; three studies). In unification defects (OR 2.28; 95% CI 1.45–3.6; *p* < 0.001; I^2^ = 0%; seven studies), there were no significant differences in second trimester miscarriage between women with unicornuate uterus (OR 2.1, 95% CI 0.95–4.61; *p* = 0.07; I^2^ = 0%; six studies), didelphys uterus (OR 1.72; 95% CI 0.6–4.9; *p* = 0.31; I^2^ = 0%; four studies), and women with a normal uterus. However, a meta-analysis of women with bicornuate uterus showed a significant increase in second trimester miscarriage (OR 2.71; 95% CI 1.4–5.23; *p* = 0.003; I^2^ = 0%; five studies).

### 3.3. Obstetric Outcomes

#### 3.3.1. Preterm Birth

Twenty-six studies reported the incidence of preterm birth in 702,769 pregnancies, consisting of 6474 pregnancies in women with CUA and 696,295 pregnancies in women with a normal uterus. The overall risk of preterm birth was significantly higher among women with CUA (OR 2.98; 95% CI 2.43–3.65; *p* < 0.001; I^2^ = 82%; Figure 3A), except for the arcuate uterus (OR 1.62; 95% CI 0.86–3.04; *p* = 0.13; I^2^ = 56%; 10 studies), compared to that in women with a normal uterus. For canalization defects (OR 3.11; 95% CI 2.24–4.32; *p* < 0.001; I^2^ = 12%; nine studies), the pooled analysis showed a significantly higher risk of preterm birth in the subseptate uterus (OR 3.15; 95% CI 1.34–7.4; *p* = 0.009; I^2^ = 61%; four studies) and septate uterus (OR 2.93; 95% CI 2.01–4.28; *p* < 0.001; I^2^ = 0%; eight studies). For unification defects (OR 3.5; 95% CI 2.74–4.46; *p* <0.001; I^2^ = 64%; 16 studies), women with unicornuate uterus (OR 2.83, 95% CI 1.92–4.19; *p* < 0.001; I^2^ = 53%; 13 studies), bicornuate uterus (OR 3.69; 95% CI 2.6–5.22; *p* < 0.001; I^2^ = 55%; 10 studies), and didelphys uterus (OR 4.93; 95% CI 3.6–6.75; *p* < 0.001; I^2^ = 13%; nine studies) had an increased rate of preterm birth compared to that in women with a normal uterus (Table 2).

#### 3.3.2. Malpresentation

Sixteen studies reported fetal malpresentation in 690,744 pregnancies, comprising 5098 pregnancies in women with CUA and 685,646 pregnancies in women with a normal uterus. The overall risk of malpresentation was significantly increased among women with CUA (OR 9.1; 95% CI 5.88–14.08; *p* < 0.001; I^2^ = 97%; Figure 3B) compared to that in women with a normal uterus. The subgroup analysis in women with arcuate uterus (OR 3.27; 95% CI 1.66–6.44; *p* < 0.001; I^2^ = 52%; seven studies) showed a significantly higher risk of malpresentation. In the case of canalization uterus (OR 11.39; 95% CI 6.24–20.78; *p* < 0.001; I^2^ = 81%; eight studies), subseptate uterus (OR 11.42; 95% CI 3.74–34.86; *p* < 0.001; I^2^ = 85%; four studies), and septate uterus (OR 11.49; 95% CI 5.24–25.17; *p* < 0.001; I^2^ = 80%; eight studies) increased the risk of malpresentation compared to that in women with a normal uterus. In unification defects (OR 8.68; 95% CI 5.82–12.95; *p* < 0.001; I^2^ = 91%; 10 studies), unicornuate uterus (OR 8.09; 95% CI 3.14–20.84; *p* < 0.001; I^2^ = 78%; eight studies), bicornuate uterus (OR 10.87; 95% CI 6.68–17.68; *p* < 0.001; I^2^ = 86%; nine studies), and didelphys uterus (OR 7.2; 95% CI 3.09–16.74; *p* < 0.001; I^2^ = 94%; nine studies) showed a higher fetal malpresentation rate (Table 2). 

#### 3.3.3. Cesarean Section

Twenty-one studies reported data on the incidence of cesarean section in a total of 698,716 pregnancies, which comprised 5283 pregnancies in women with CUA and 693,433 pregnancies in women with a normal uterus. A significant difference was observed in the incidence of cesarean section among women with all CUA (OR, 2.87; 95% CI, 1.56–5.26; *p* < 0.001; I^2^ = 99%; Figure 3C), except for unicornuate uterus (OR 1.24; 95% CI 0.76–2.03; *p* = 0.39; I^2^ = 71%; nine studies), compared to that in women with a normal uterus. The outcomes of subgroup analysis (Table 2) showed a significantly higher risk of cesarean section in the arcuate uterus (OR 2.22; 95% CI 1.07–4.61; *p* = 0.03; I^2^ = 61%; six studies), subseptate uterus (OR 5.91; 95% CI 1.59–21.95; *p* = 0.008; I^2^ = 87%; three studies), septate uterus (OR 4.84; 95% CI 2.33–10.02; *p* < 0.001; I^2^ = 68%; six studies), bicornuate uterus (OR 5.23; 95% CI 2.11–12.96; *p* < 0.001; I^2^ = 95%; eight studies), and didelphys uterus (OR 7.55; 95% CI 2.4–23.72; *p* < 0.001; I^2^ = 96%; seven studies).

#### 3.3.4. Placental Abruption

Nine studies reported data on the incidence of placental abruption in a total of 618,188 pregnancies, consisting of 3145 pregnancies in women with CUA and 615,043 pregnancies in women with a normal uterus. The overall risk of placental abruption was significantly higher among women with CUA (OR 3.12; 95% CI, 1.58–6.18; *p* = 0.001; I^2^ = 84%; Figure 3D) than that in women with a normal uterus. In the subgroup analysis based on CUA subtype (Table 2), it was shown that there was a significantly higher risk of placental abruption in the arcuate uterus (OR 4.56; 95% CI 1.03–20.06; *p* = 0.04; I^2^ = 36%; four studies), septate uterus (OR 5.33; 95% CI 1.5–18.95; *p* = 0.01; I^2^ = 0%; three studies), unicornuate uterus (OR 7.78; 95% CI 1.99–30.45; *p* = 0.003; I^2^ = 0%; three studies), and bicornuate uterus (OR 6.53; 95% CI 1.96–21.78; *p* = 0.002; I^2^ = 70%; four studies), but not in the didelphys uterus (OR 2.68; 95% CI 0.51–14.17; *p* = 0.24; I^2^ = 0%; three studies).

### 3.4. Neonatal Outcomes

#### 3.4.1. Intrauterine Growth Restriction or Small for Gestational Age

Ten studies reported the incidence of IUGR or SGA in 617,231 pregnancies, consisting of 3666 pregnancies in women with CUA and 613,565 pregnancies in women with a normal uterus. Meta-analysis of ten studies revealed that women with CUA have an increased risk of IUGR or SGA compared to that in women with a normal uterus (OR 2.53; 95% CI 1.77–3.62; *p* < 0.001; I^2^ = 82%; Figure 4A). In the subgroup analysis (Table 2), no significant differences in the rates of IUGR or SGA were observed in women with arcuate uterus (OR 3.77; 95% CI 0.92–15.46; *p* = 0.07; I^2^ = 75%; five studies) and septate uterus (OR 1.9; 95% CI 0.89–4.08; *p* = 0.1; I^2^ = 2%; five studies). The outcome of the subgroup analysis showed that the risk of IUGR or SGA was significantly higher in women with subseptate uterus (OR 2.4; 95% CI 1.13–5.09; *p* = 0.02; I^2^ = 0%; three studies), unicornuate uterus (OR 3.5; 95% CI 1.24–9.91; *p* = 0.02; I^2^ = 43%; five studies), bicornuate uterus (OR 2.84; 95% CI 1.68–4.8; *p* < 0.001; I^2^ = 57%; six studies), and didelphys uterus (OR 4.03; 95% CI 2.0–8.12; *p* < 0.001; I^2^ = 44%; six studies) than that in women with a normal uterus.

#### 3.4.2. Low Birth Weight

Nine studies reported data on the incidence of LBW in 266,765 pregnancies, consisting of 2239 pregnancies in women with CUA and 264,526 pregnancies in women with a normal uterus. The result for LBW was not significant (OR 1.59; 95% CI 0.94–2.68; *p* = 0.08; I^2^ = 93%; Figure 4B) in women with CUA compared to women with a normal uterus. In the subgroup analysis by CUA subtype (Table 2), no significant differences were found between LBW rate in women with arcuate uterus (OR, 1.5; 95% CI 0.7–3.25; *p* = 0.3; I^2^ = 0%; two studies) or septate uterus (OR 1.73; 95% CI 0.91–3.29; *p* = 0.09; I^2^ = 0%; two studies) and that in women with a normal uterus. The risk of LBW was a higher in women with unification defects (OR 1.99; 95% CI 1.38–2.87; *p* < 0.001; I^2^ = 57%; seven studies), with an OR of 1.8 in unicornuate uterus (OR 1.84; 95% CI 1.08–3.14; *p* = 0.02; I^2^ = 72%; seven studies), 1.9 in bicornuate uterus (OR 1.91; 95% CI 1.12–3.27; *p* = 0.02; I^2^ = 0%; two studies), and 2.9 in didelphys uterus (OR 2.87; 95% CI 1.38–5.97; *p* = 0.005; I^2^ = 0%; two studies).

#### 3.4.3. Perinatal Mortality

Twelve studies reported data on perinatal mortality in a total of 617,278 pregnancies, consisting of 4610 pregnancies in women with CUA and 612,668 pregnancies in women with a normal uterus. The overall risk of perinatal mortality was higher in women with CUA than that in women with a normal uterus (OR, 2.17; 95% CI 1.46–3.23; *p* < 0.001; I^2^ = 48%; Figure 4C). In the subgroup analysis by CUA subtype (Table 2), no significant differences were observed in perinatal mortality rate in women with an arcuate uterus (OR 2.11; 95% CI 0.79–5.63; *p* = 0.14; I^2^ = 0%; five studies), subseptate uterus (OR 2.51; 95% CI 0.82–7.69; *p* = 0.11; I^2^ = 0%; three studies), and didelphys uterus (OR 1.75; 95% CI 0.77–3.96; *p* = 0.18; I^2^ = 0%; five studies). Conversely, women with a septate uterus (OR 2.57; 95% CI 1.08–6.08; *p* = 0.03; I^2^ = 0%; six studies), unicornuate uterus (OR 3.85; 95% CI 1.61–9.2; *p* = 0.002; I^2^ = 42%; nine studies), and bicornuate uterus (OR 3.17; 95% CI 2.08–4.84; *p* < 0.001; I^2^ = 0%; seven studies) showed a higher rate of perinatal mortality than that in those with a normal uterus. 

### 3.5. Other Outcomes

The pooled analysis showed that women with CUA had a higher rate of preeclampsia (OR 1.25; 95% CI 1.07–1.46; *p* = 0.004; I^2^ = 0%; five studies), IUFD (OR 2.06; 95% CI 1.36–3.11; *p* < 0.001; I^2^ = 7%; seven studies), PPROM (OR 3.5; 95% CI 2.22–5.54; *p* < 0.001; I^2^ = 26%; three studies), recurrent pregnancy loss (OR 2.61; 95% CI 2.31–2.94; *p* < 0.001; I^2^ = 0%; three studies), and cervical incompetence (OR 7.94; 95% CI 3.81–16.55; *p* < 0.001; I^2^ = 85%; four studies) than that in women with a normal uterus. No significant differences in the rate of ectopic pregnancy (OR 1.28; 95% CI 0.81–2.02; *p* = 0.29; I^2^ = 0%; 11 studies), postpartum hemorrhage (OR 1.02; 95% CI 0.67–1.55; *p* = 0.93; I^2^ = 0%; six studies), and placenta previa (OR 1.56; 95% CI 0.6–4.07; *p* = 0.36; I^2^ = 83%; six studies) were found between women with CUA and those with a normal uterus. The outcomes from the subgroup analysis according to the CUA subtype are shown in Table 2. Concerning ectopic pregnancy, preeclampsia, and placenta previa, no significant differences were noted in the arcuate uterus and canalization defects. In the unification defects, the rate of placenta previa was significantly higher in women with a bicornuate uterus (OR 3.59; 95% CI 1.89–6.82; *p* <0.001; I^2^ = 0%; four studies) than that in women with a normal uterus. Additionally, the overall risk of PPROM (OR 4.66; 95% CI 2.83–7.69; *p* < 0.001; I^2^ = 0%; two studies), and cervical incompetence (OR 9.04; 95% CI 6.11–13.37; *p* < 0.001; I^2^ = 0%; three studies) was increased in women with unification defects. Likewise, women with septate uterus showed a higher rate of PPROM (OR 4.66; 95% CI 1.79–12.15; *p* = 0.002; I^2^ = 0%; two studies) than that in women with a normal uterus. In women with bicornuate uterus, the risk of recurrent pregnancy loss (OR 2.69; 95% CI 2.05–3.52; *p* < 0.001; I^2^ = 0%; two studies) increased, and in women with unicornuate uterus, the risk of IUFD (OR 2.4; 95% CI 1.27–4.53; *p* = 0.007; I^2^ = 0%; four studies) increased.

## 4. Discussion

This systematic review and meta-analysis investigated the reproductive, obstetric, and neonatal outcomes of women with CUA. This extensive and updated meta-analysis is an attempt to review all published studies on the reproductive effects of CUA, to investigate the association between CUA and all obstetric complications analyzed in the studies. Based on the results of this meta-analysis, the presence of CUA has a negative impact on most pregnancy outcomes, further supporting the theory of inadequate implantation, fetal development, and pregnancy maintenance in CUA. 

We found a significant decrease in the clinical pregnancy rate in the septate uterus and unicornuate uterus, over threefold increased risk of miscarriage in women with canalization defects, and a twofold increased risk in the bicornuate uterus, but this risk of miscarriage was not found in the unicornuate and didelphys uterus. The risk of preterm birth in women with both canalization and unification defects was over threefold higher than that in women with a normal uterus. As gestational weeks progressed, malpresentation and, consequently, cesarean section rates were significantly higher in all subtypes of CUA due to a lack of uterine volume corresponding to fetal growth in the second trimester of pregnancy. Particularly, in all subtypes of CUA, the risk of placental abruption was ninefold higher in the canalization defects, sevenfold higher in the unification defects, and fivefold higher even in the arcuate uterus. Additionally, we found that unification defects can be associated with a range of obstetric and fetal complications, including LBW (twofold risk), IUGR (over threefold risk), IUFD (2.5-fold risk), perinatal mortality (threefold risk), placenta previa (over threefold risk), and cervical incompetence (ninefold risk). We noted a similar risk of PPROM has been shown in the septate uterus and unification defects, with a fivefold increase. A septate uterus was not associated with adverse neonatal outcomes, including LBW, IUGR or SGA, and IUFD, except for perinatal mortality (threefold risk). There were no significant differences in preeclampsia and ectopic pregnancy between the CUA subtypes and a normal uterus. Unlike other CUA subtypes, there was no significant difference in most pregnancy outcomes between women with an arcuate uterus and those with a normal uterus, except for second trimester miscarriage, placental abruption, malpresentation, and cesarean section. An arcuate uterus, milder forms of CUA, or normal variants are less likely to influence fertilization and early implantation processes, and impact the risk of miscarriage. The more severe forms of CUA affect all stages of reproduction, from fertilization to early and late pregnancy complications. 

The causes and pathophysiological mechanisms of various reproductive and obstetrical complications in patients with CUA remain unclear. Embryo implantation is influenced by the vascularity and thickness of the endometrium and the morphology of the uterine cavity [60]. The uterine septum is known to be composed of fibroelastic tissue with inadequate vascularization of the endometrium, which reduces endometrial receptivity to estrogen, resulting in reduced endometrial maturation and proliferation [61,62]. Miscarriage can easily occur as the embryos implant on the septum because the septum with suboptimal endometrium is morbid with reduced blood supply. Even if implantation occurs, it does not provide a proper environment for subsequent placental and embryonic growth. In contrast, Dabirashrafi et al. found significantly more blood vessels and muscle tissues and less connective tissue in the septum of the uterus, which might result in poor decidualization, placentation, and uncoordinated muscle contractility [63]. Abnormalities of space in the uterine cavity, arrangement of uterine musculature, and impaired ability to distend are likely to have a negative effect on pregnancy maintenance. In addition, increased muscle mass and decreased connective tissue in the malformed cervix can cause asymmetric uterine cavity pressure, impairing the ability of distention and growth of the uterine cavity, which also leads to late miscarriage and preterm birth. 

Currently, there is no universally accepted standard classification for CUA [11]. The differences in sensitivity and specificity of the diagnostic tests inevitably affect the classification and diagnosis of CUA subtypes. Therefore, classification systems and standardized tests are essential for counseling and determining the management of CUA patients. Accurate assessment of the internal and external uterine contours is critical for diagnosing and classifying CUA. Previously, the gold standard diagnostic method was a combination of laparoscopy and hysteroscopy. However, imaging techniques are less invasive, such as 2D- or 3D-US, HSG, sonohysterography, and magnetic resonance imaging (MRI) for screening, diagnosing, and classifying CUA [64]. Particularly, 2D-US and HSG are helpful in screening for CUA, while 3D-US and MRI are suitable for categorizing CUA accurately [40,65,66,67]. 

There is still insufficient evidence on the efficacy and safety of surgical interventions for improving reproductive performance. There is a lack of evidence on improving reproductive outcomes with surgical intervention. Some reports have shown that hysteroscopic septal division can reduce the risk of miscarriage and improve live birth rates [6,68,69,70], but surgical management for incidentally diagnosed septum and fusion or unification defects is controversial and unproven [71]. Various hysteroscopic instruments, including microscissors, forceps, operating loop, electrosurgical needle, and laser energy, have been proposed for surgical correction [72]. Gradually, several methods are being introduced to help the proliferation of endometrial tissue, reduce adhesion formation and minimize intraoperative or postoperative complications [73]. Tissue vaporization and coagulation with low power diode laser energy can rapidly remove the uterine septum without damaging the underlying myometrium [74,75]. Still, there is a lack of large scale studies on which technique is better.

Chan et al. [18] previously reported the correlation between CUA and representative pregnancy outcomes, such as miscarriage, preterm birth, and malpresentation, by meta-analyzing nine studies. Venetis et al. [19] analyzed the pregnancy outcomes in women with CUA by adding LBW, IUGR, PPROM, placental abruption, and perinatal mortality extracted from 24 studies. We conducted an extensive and updated meta-analysis to evaluate the association between the different types of CUA and various reproductive, obstetric, and neonatal outcomes, in 37 studies where a more comprehensive assessment of the evidence was available. Undoubtedly, the negative impacts of CUA on pregnancy begin in the early stages of fertilization and the impacts span the entire reproductive process. However, the lack of data on individual CUA subtypes makes it difficult to draw clear conclusions about the exact subtype specific effects on complications and their underlying mechanisms. Although most women with CUA experience a normal reproductive outcome, it is necessary to advise on the risks that may be increased depending on the type and severity of CUA.

To fully investigate the effects of CUA on obstetric and fetal complications, collecting all published studies and conducting a thorough and conclusive meta-analysis may be hampered by the heterogeneity of the currently available studies. This meta-analysis was confounded by the heterogeneity of the included clinical studies. The use of several diagnostic methods with variable accuracies, the absence of a universal classification system, discrepancies in interpretation of CUA classification, and the heterogeneity of study populations have all contributed to a difficulty in deriving the study results. As we collected and analyzed studies published over the past half century, there are inevitable differences in study population characteristics, diagnostic methods, classification system of CUA, and definition of outcome. Most of the included studies have a retrospective design, and various confounding variables, such as age, body mass index, hormonal status, and socioeconomic status, that can introduce bias. Therefore, future studies should be conducted through well designed prospective observational studies that consider these potential confounding factors to establish more detailed evidence of the risk of adverse reproductive outcomes in women with CUA. 

This study showed that all subtypes of CUA have an increased risk of second trimester miscarriage, preterm birth, placental abruption, fetal malpresentation, and cesarean section. It was found that women with canalization defects seem to have the poorest reproductive performance in early pregnancy. In unification defects, only the unicornuate uterus is shown to reduce fertility, and bicornuate uterus increases miscarriage, while didelphys uterus is not associated with risk of fertility and miscarriage. However, unification defects were primarily associated with obstetric and neonatal outcomes throughout pregnancy. The abnormal uterine environment disrupts fertilization, implantation, and later pregnancy and birth outcomes. In conclusion, women with CUA can develop adverse reproductive, obstetric, and neonatal outcomes according to CUA subtypes. The uterus is an essential organ in which critical phenomena of the reproductive process occur, including sperm migration, embryo implantation, fetal development and growth, and, finally, the induction of labor and childbirth. CUA can influence these uterine functions and prevent successful pregnancies. It may be beneficial for clinicians to care for women with CUA and inform them about potential complications and treatment options through evidence based counseling and an accurate diagnosis before and during pregnancy. 

## Figures and Tables

**Figure 1 jcm-10-04797-f001:**
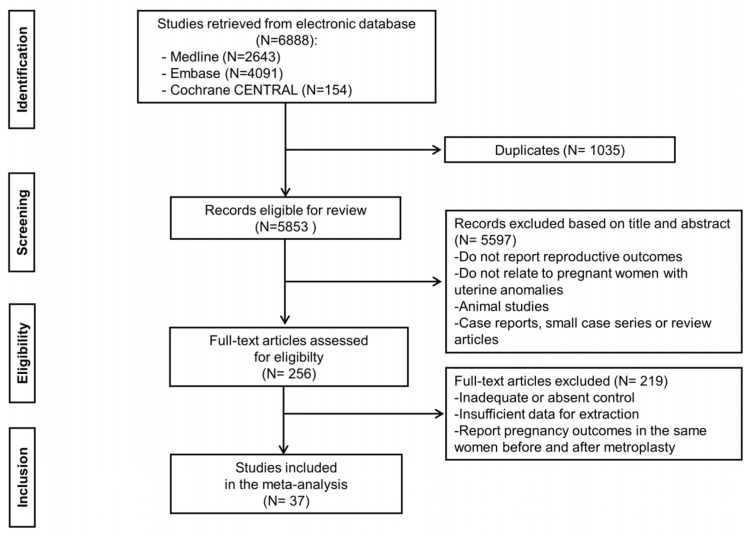
Flow diagram of the study selection process.

**Figure 2 jcm-10-04797-f002:**
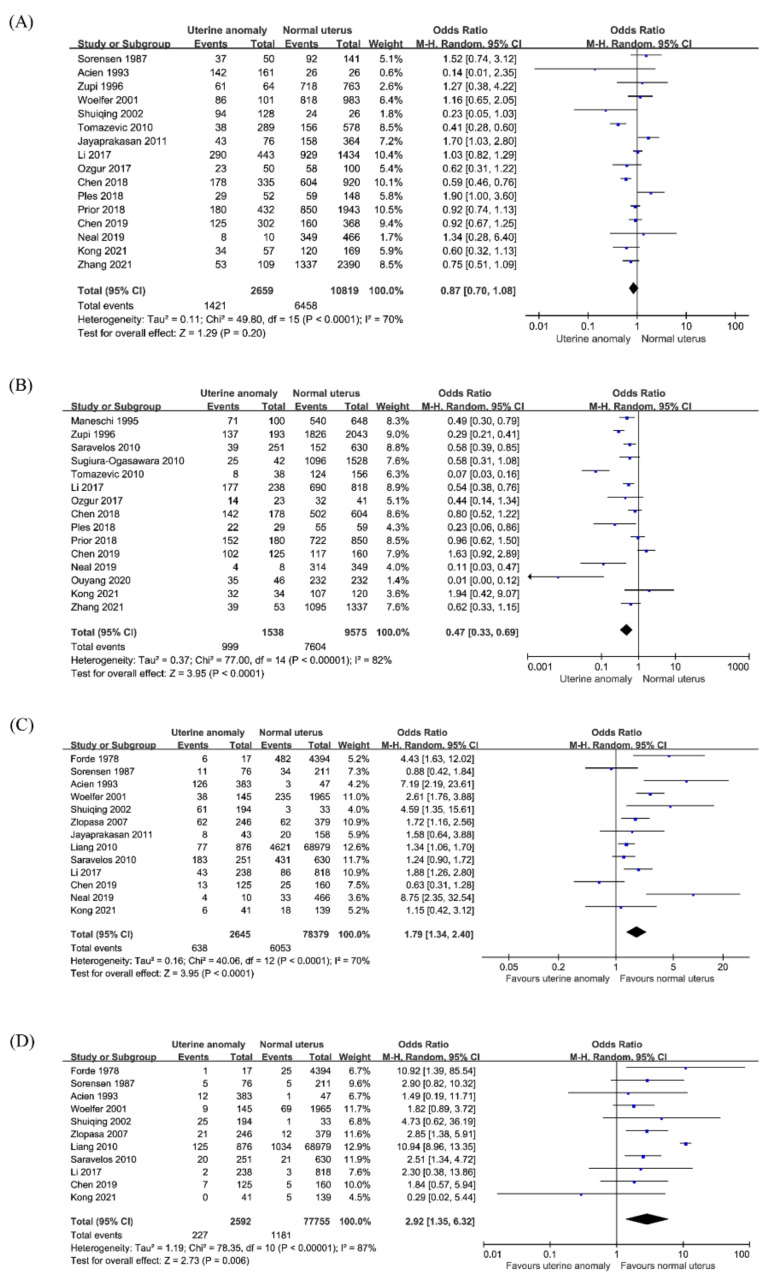
Reproductive outcomes in women with congenital uterine anomalies. (**A**) Clinical pregnancy rate, (**B**) live birth, (**C**) first trimester miscarriage, (**D**) second trimester miscarriage.

**Figure 3 jcm-10-04797-f003:**
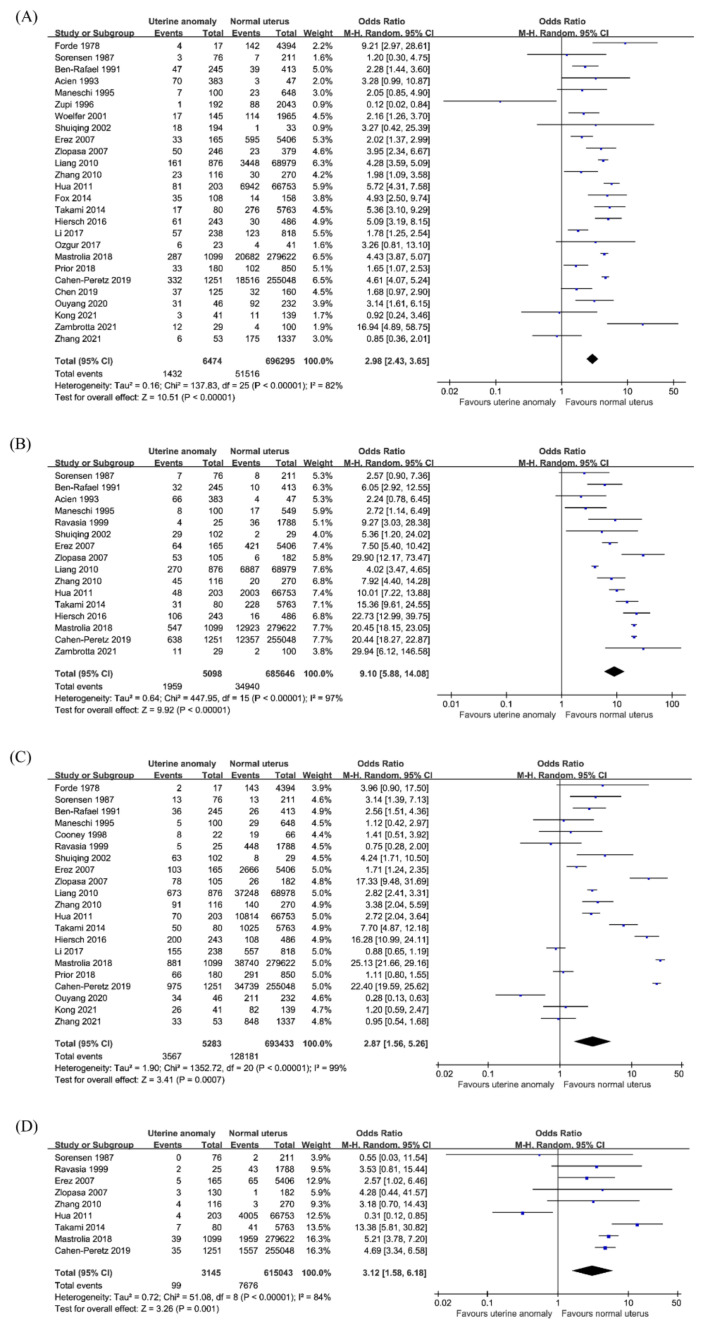
Obstetric outcomes in women with congenital uterine anomalies. (**A**) Preterm birth, (**B**) malpresentation, (**C**) cesarean section, (**D**) placental abruption.

**Figure 4 jcm-10-04797-f004:**
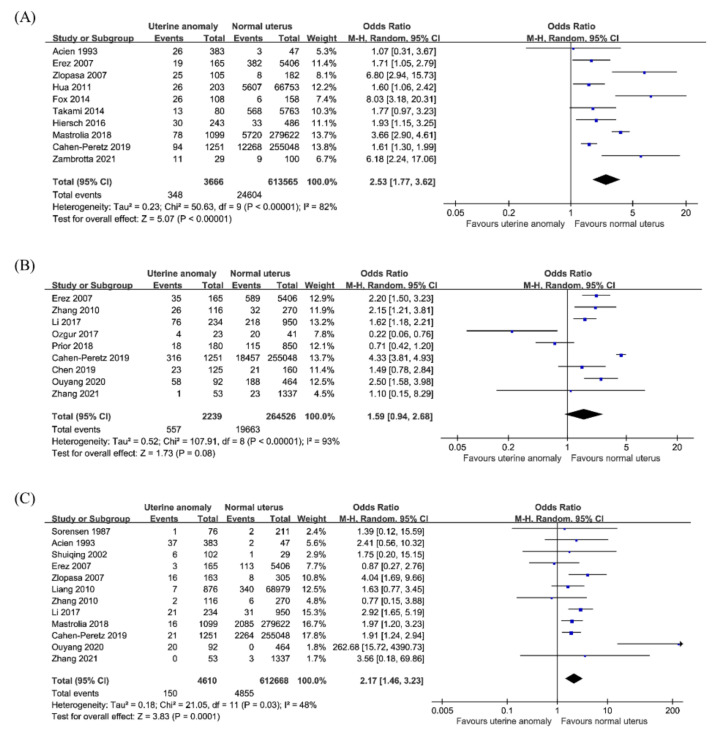
Neonatal outcomes in women with congenital uterine anomalies. (**A**) Intrauterine growth restriction or small for gestational age, (**B**) low birth weight, (**C**) perinatal mortality.

**Table 1 jcm-10-04797-t001:** Characteristics of included studies.

Author & Year	Type of Study	Population	Exclusion Criteria	Mode of Conception	Women with CUA	Women with a Normal Uterus	CUA Types	Mode of Diagnosis	Method of Classification of Anomalies
Forde, 1978	Retrospective	All deliveries	Not reported	Not reported	*n* = 17 deliveries	*n* = 4394 deliveries	Arcuate, bicornuate, didelphys	Hysterography	Jarcho (1946)
Sorensen and Trauelsen, 1987	Retrospective	Infertile women	Women that could not be traced and women with major uterine anomalies	Not reported	*n* = 50	*n* = 141	Arcuate	HSG	According to the degrees of fundal excavation, if H/L ratio ≥ 0.1, minor Müllerian anomalies were diagnosed. (H: the distance from the nadir of the fundal indentation to the line connecting the summits of the uterine horns L: the length of this line)
Ben-Rafael, 1991	Retrospective	(A) Infertile women or (B) women who have experienced recurrent fetal loss	Women with arcuate uterus or subseptate uterus	Not reported	(A) *n* = 27 (B) *n* = 40	(A) *n* = 89 (B) *n* = 41	Unicornuate, bicornuate, didelphys	HSG	Not reported
Acien, 1993	Retrospective and prospective	Women with uterine or genitourinary anomalies	Patients with Rokitansky syndrome, hypoplastic uterus, and incomplete case studies	Not reported	Uterine or uterovaginal malformations, *n* = 176	Other genital and/or urinary anomalies, but normal uterus, *n* = 28	Arcuate, unicornuate, bicornuate, septate, didelphys	Clinical examination, ultrasound, HSG, pyelography	The criteria of Jarcho (1946), Buttram and Gibbons (1979), American Fertility Society (1988)
Maneschi, 1995	Prospective	All women with abnormal uterine bleeding	Women who had never attempted to conceive or filled the incomplete questionnaire, women with a submucosal myoma or unknown shape of the uterine cavity at the time of pregnancy	Not reported	*n* = 33 (arcuate:21, sepate/bicornuate:12)	*n* = 216	Arcuate, septate/bicornuate (not differentiated)	Hysteroscopy	Arcuate uterus: fundal protrusion of <20% of the uterine cavity
Zupi, 1996	Retrospective	Patients who underwent outpatient hysteroscopy for reasons other than infertility	Women who did not know the state of their uterine cavity at conception, women who had undergone voluntary abortion, women with submucous fibroids or synechiae	Not reported	*n* = 64 (arcuate:49, sepate/bicornuate:13, unicornuate: 2)	*n* = 763	Arcuate, septate/bicornuate, unicornuate	Hysteroscopy	Septate or bicornuate uterus: a double cavity, separated by a mid-cavity septum that covered at least one third of the uterine cavity Arcuate uterus: a uterus with a fundal notch consisting of less than one third of the cavity (Fedele et al. 1990)
Cooney, 1998	Retrospective	All pregnancies with fetal cardiac activity in the first trimester	Equivocal images and a lack of independent confirmation	Not reported	*n* = 22	*n* = 66	Uterine duplication anomaly (bicornuate, septate, didelphys)	Surgical procedure before or after the sonogram, hysteroscopy	Not reported
Ravasia, 1999	Retrospective	Women undergoing trial of labor after previous caesarean section	Not reported.	Not reported.	*n* = 25	*n* = 1788	Unicornuate, bicornuate, septate, didelphys	Laparoscopy, hysteroscopy after an abnormal result of US or HSG, during first cesarean delivery	Not reported
Woelfer, 2001	Prospective	Women referred for gynecologic examination	Ongoing pregnancy, history of infertility or recurrent miscarriage, presence of uterine fibroids that distorted the uterine cavity, and previous hysterectomy or myomectomy.	Not reported.	*n* = 106	*n* = 983	Arcuate, subseptate, bicornuate	2D- & 3D-US	American Fertility Society (1988)
Shuiqing, 2002	Retrospective	Women with CUA and women with other urinary or genital anomalies but with normal uterus	Not reported	Not reported	*n* = 153	*n* = 27	Unicornuate, bicornuate, septate, didelphys	History, clinical examination, high resolution US, HSG, hysteroscopy, laparoscopy, laparotomy	Buttram (1983)
Erez, 2007	Retrospective	All patients attempted vaginal birth after cesarean section after previous cesarean section	Multiple pregnancies, more than one previous cesarean section and known congenital and/or chromosomal fetal anomalies	Not reported	*n* = 165	*n* = 5406	Arcuate, unicornuate, bicornuate, septate, didelphys	All the surgery reports of the primary cesarean section	American Fertility Society (1988)
Zlopasa, 2007	Retrospective	All pregnant women	Twin gestation, chorioamnionitis, presence of submucosal myomas, fetal chromosomopathy, maternal diabetes, or fertilization in vitro	Exclude IVF	*n* = 130	*n* = 182	Arcuate, unicornuate, bicornuate, subseptate, septate, didelphys	Previous surgery, sonohysterographic evaluation, laparoscopy with hysterography, or hysteroscopy	American Fertility Society (1988)
Ban-Frangez, 2009	Retrospective	Women with a singleton intrauterine pregnancy with fetal heart beat after IVF or ICSI	Extrauterine pregnancies, multiple pregnancies and cases with an empty gestational sac	IVF/ICSI	*n* = 31	*n* = 62	Arcuate, septate	2D-US without intrauterine saline infusion, hysteroscopy	American Fertility Society (1988)
Liang, 2010	Retrospective	All pregnant women (from week 28 of pregnancy to day 7 postdelivery)	Women with medical complications and history of cesarean delivery	Not reported	*n* = 715	*n* = 68,979	Unicornuate, bicornuate, incomplete septate, complete septate, didelphys	Patient’s history of illness, physical examination, high resolution US, HSG, hysteroscopy, laparoscopy, laparotomy	Buttram and Gibbons (1979)
Saravelos, 2010	Retrospective	Women who have experienced recurrent miscarriages	Patients who had received medical treatment (e.g., low molecular weight heparin, acetyl-salicylic acid, steroids) or surgery (e.g., septotomy, Strassman’s metroplasty, cervical cerclage)	Not reported	*n* = 56	*n* = 107	Arcuate, unicornuate, bicornuate, septate, didelphys	2D-US, HSG, hysteroscopy, laparoscopy	American Fertility Society (1988)
Sugiura-Ogasawara, 2010	Retrospective	Patients with a history of two or more consecutive miscarriages	Patients with structural chromosome abnormalities.	Not reported	*n* = 42	*n* = 1528	Unicornuate, bicornuate, septate, didelphys	Laparoscopy, laparotomy and/or magnetic resonance imaging	American Fertility Society (1988) & Tompkin’s index
Tomazevic, 2010	Retrospective	Infertile women who underwent embryo transfers in IVF/ICSI cycles	Not reported	IVF/ICSI	Embryo transfers before hysteroscopic resection, *n* = 289	Embryo transfers, *n* = 578	Arcuate, subseptate, septate	2D-US without intrauterine saline infusion, hysteroscopy	American Fertility Society (1988) If the uterine septum measured 1.3~1.5 cm in length, it was defined as arcuate uterus.
Zhang, 2010	Retrospective	All patients referred for delivery	Multiple pregnancies, congenital and/or chromosomal fetal anomalies	Infertility treatment or natural cycle	*n* = 116	*n* = 270	Arcuate, unicornuate, bicornuate, septate, didelphys	Previous surgery or sonohysterography, laparoscopy with hysterography, hysteroscopy	American Fertility Society (1988)
Hua, 2011	Retrospective	all consecutive singleton pregnancies undergoing routine anatomic survey	Patients with incomplete follow-up data	Not reported	*n* = 203	*n* = 66,753	Unicornuate, bicornuate, septate, didelphys	Not reported	Not reported
Jayaprakasan, 2011	Prospective	Women who have undergone IVF/ICSI	Women who have one or more uterine fibroids or polyps distorting the endometrial cavity or if the ultrasound view was unclear and not good enough to allow a definitive diagnosis to be made	IVF/ICSI	*n* = 76	*n* = 364	Arcuate, unicornuate, bicornuate, subseptate, septate, T-shaped uterus	2D-, 3D-US	Modified American Fertility Society classification proposed by Salim et al. (2003)
Fox, 2014	Retrospective	Women with singleton pregnancies ≥ 22 weeks delivered	Not reported	ART & natural cycle	*n* = 158	*n* = 158	Arcuate, unicornuate, bicornuate, septate, didelphys, T-shaped uterus	Saline infusionsonohysterogram, MRI, hysteroscopy, laparoscopy	American Fertility Society (1988)
Takami, 2014	Retrospective	Women who delivered a live singleton baby after 22 gestational weeks	Women whose fetuses had congenital anomalies or with a history of surgery	Infertility treatment or natural cycle	*n* = 80	*n* = 5763	Unicornuate, bicornuate, subseptate, incomplete septate, complete septate, didelphys	US, internal examinations during pregnancy, findings at cesarean section, MRI, HSG	American Fertility Society (1988)
Hiersch, 2016	Retrospective	Women who delivered at or beyond 24 weeks of gestation	Women who underwent any surgical treatment of Müllerian anomalies, pregnancies with uncertain pregnancy dating or those complicated by stillbirth or major fetal anomalies	IVF, IUI or natural cycle	*n* = 243	*n* = 486	Unicornuate, bicornuate, septate, didelphys	HSG, US, hysterosonography, CT, MRI, hysteroscopy, laparoscopy, laparotomy	American Fertility Society (1988)
Li, 2017	Retrospective	Infertile women who achieved pregnancy with IVF cycles	Patients with donor oocytes, preimplantation genetic diagnosis, preimplantation genetic screening, parental chromosomal abnormalities, spontaneous/selective reduction, triplet pregnancies, induced labor for congenital fetal structural or chromosomal abnormalities, uterine fibroids or polyps distorting the endometrial cavity	IVF-ET	*n* = 238	*n* = 818	Unicornuate	3D-US, HSG, hysteroscopy, laparoscopy	ESHRE/ESGE classification system
Mastrolia, 2017	Retrospective	All women carrying a singleton pregnancy who delivered	Patients with multiple pregnancies or missing data	ART & natural cycle	*n* = 444	*n* = 279,662	Bicornuate	Workup for infertility or recurrent pregnancy loss, during pregnancy, or at the time of cesarean delivery	American Fertility Society (1988)
Ozgur, 2017	Retrospective	Patients who had first infertility consultations/ICSI with fresh and cryopreserved embryo transfer	Patients with intrauterine abnormalities or who underwent therapeutic surgery	IVF/ICSI	*n* = 50	*n* = 100	Unicornuate	2D-US, HSG, saline-infused sonography, hysteroscopy, laparoscopy	American Fertility Society (1988)
Chen, 2018	Retrospective	Women who underwent IVF/ICSI cycles	Other uterine malformations (septum, unicornuate uterus class IVa, bicornuate uterus), endometrial lesions (polyps, endometrial hyperplasia, intrauterine adhesions), uterine fibroids distorting uterine cavity diagnosed by TVS or hysteroscopy, sonographic features of adenomyosis, chromosomal abnormality of male or female Partner, patients who undertook a donor oocyte program or had preimplantation genetic diagnosis/preimplantation genetic screening, patients who had cancelled IVF cycle that did not result in embryo transfer	IVF/ICSI	*n* = 342	*n* = 1026	Unicornuate	3D-US, hysteroscopy, laparoscopy, MRI	ESHRE/ESGE classification system
Mastrolia, 2018	Retrospective	All women who have delivered	Patients with multiple pregnancies or missing data	ART & natural cycle	*n* = 1099	*n* = 279,662	All CUA	Workup for infertility or recurrent pregnancy loss, during pregnancy or at the time of cesarean delivery	American Fertility Society (1988)
Ples, 2018	Retrospective	Patients with infertility, who underwent ART	Patients with associated uterine pathology one or more polyps, synechiae, or submucosal myoma, patients in whom the ultrasound image was not sufficient for a definitive diagnosis	ART	*n* = 52	*n* = 148	Dysmorphic uterus, septate, Bicorporeal uterus, aplastic uterus	2D & 3D-US	ESHRE/ESGE classification system
Prior, 2018	Prospective	Women with subfertility, defined as failure to conceive after regular unprotected sexual intercourse for 2 years, who underwent ART	The presence of fibroids, intrauterine device or polyps distorting the cavity, Asherman’s syndrome, previous hysteroscopic surgery and/or poor quality of images	IVF/ICSI	*n* = 432	*n* = 1943	Arcuate, unicornuate, bicornuate, subseptate, septate, didelphys	2D & 3D-US	Modified American Fertility Society classification proposed by Salim et al. (2003)
Cahen-Peretz, 2019	Retrospective	All women who delivered	Multifetal pregnancies, unknown gestational age, gestational age of less than 24 weeks upon delivery, fetuses with congenital malformations	ART & natural cycle	*n* = 1251	*n* = 255,048	Arcuate, unicornuate, bicornuate, septate, didelphys	US, HSG, hysterosonography, MRI, hysteroscopy, laparoscopy, laparotomy	Not reported
Chen, 2019	Retrospective	Patients receiving IVF/ICSI	Oocyte donor treatment cycles, abnormal uterine bleeding, endometrial fibroids or polyps, intrauterine adhesion, premature ovary insufficiency, polycystic ovary syndrome, consecutive spontaneous abortion history ≥ three times	IVF/ICSI	*n* = 160	*n* = 160	Unicornuate	TVS, HSG, hysteroscopy, laparoscopy	Not reported
Neal, 2019	Prospective	Infertile patients planning to undergo a single thawed euploid blastocyst transfer	Use of a gestational carrier, body mass index 40 kg/m^2^ or over, previously diagnosed uterine anomalies of the Class U2–U5 variety, history of myomectomy, communicating hydrosalpinx	IVF-ET	*n* = 10	*n* = 472	T-shaped uterus	3D-US	ESHRE/ESGE classification system
Ouyang, 2020	Retrospective	All patients successfully achieved first pregnancies and delivered at ≥22 weeks after IVF-ET	Maternal age ≥ 40 years old, body mass index outside the range of 18–28, only one ovary detected, uterine fibroids or polyps distorting the endometrial cavity, received donor oocytes, preimplantation genetic diagnosis, preimplantation genetic screening, parental chromosomal abnormalities, spontaneous reduction, monochorionic twin or triplet pregnancies, early or late miscarriage, ectopic pregnancy, induced labour, abnormal chromosomal karyotypes, other urinary tract malformations	IVF/ICSI	*n* = 206	*n* = 314	Unicornuate	3D-US, HSG, hysteroscopy, laparoscopy, laparotomy	ESHRE/ESGE classification system
Kong, 2021	Retrospective	Women who underwent first IVF/ICSI cycles	Severe systemic disease, presence of uterine or pelvic disease, such as severe intrauterine adhesions, uterine adenomyosis, or untreated hydrosalpinx, presence of a chromosomal abnormality in the male or female partner, participation in a donor oocyte program or presence of a preimplantation genetic test, giving up treatment midway or no follow up information availability	IVF/ICSI	*n* = 58	*n* = 174	Bicornuate	Hysteroscopy combined with laparoscopy surgery or cesarean section, MRI, 3D-US, 2D-US combined with hysteroscopy and/or HSG under X-ray	American Fertility Society (1988)
Zambrotta, 2021	Retrospective	Women with a history of one or more pregnancies or current pregnancy with diagnosis of uterine anomalies	Women who have never had confirmed pregnancy by beta-HGC with first trimester ultrasound and who did not have specific diagnosis of uterine malformations, who underwent ART cycles and who experienced at least one miscarriage	Natural cycle	*n* = 29	*n* = 100	Arcuate, unicornuate, bicornuate, incomplete septate, complete septate, didelphys	US	American Fertility Society (1988)
Zhang, 2021	Retrospective	Infertile women who underwent IVF/ICSI	Other types of uterine malformations (mediastinal uterus, bicorne uterus, etc., diminished ovarian reserve, endometrial lesions, uterine fibroids, adenomyosis, polycystic ovary syndrome, recurrent miscarriage, chromosomal abnormalities	IVF/ICSI	*n* = 109	*n* = 2390	Unicornuate	TVS, HSG, hysteroscopy, laparoscopy	Not reported

CUA, congenital uterine anomalies; 2D-US, two dimensional ultrasound; 3D-US, three dimensional ultrasound; HSG, hysterosalpingography; TVS, transvaginal sonography, MRI, magnetic resonance imaging; IVF, in vitro fertilization; ET, embryo transfer; ICSI, intracytoplasmic sperm injection; ART, assisted reproductive technology.

**Table 2 jcm-10-04797-t002:** Reproductive, obstetric, and neonatal outcomes according to the subtype of congenital uterine anomalies.

	All CUA	Arcuate Uterus	Canalization Defects			Unification Defects			
			Subseptate	Septate	All	Unicornuate	Bicornuate	Didelphys	All
Clinical pregnancy rate	0.87 (0.70, 1.08)	1.00 (0.64, 1.58)	0.73 (0.28, 1.92)	0.45 (0.21, 0.95)	0.59 (0.32, 1.08)	0.75 (0.58, 0.99)	0.57 (0.32, 1.03)	0.36 (0.09, 1.39)	0.72 (0.57, 0.90)
Live births	0.47 (0.33, 0.69)	0.45 (0.22, 0.92)	0.18 (0.02, 1.27) *	0.25 (0.09, 0.75)	0.24 (0.10, 0.57)	0.57 (0.34, 0.96)	0.61 (0.36, 1.02)	0.15 (0.01, 2.56) *	0.60 (0.40, 0.90)
First trimester miscarriage	1.79 (1.34, 2.40)	1.38 (0.88, 2.17)	4.36 (2.64, 7.21)	2.55 (1.33, 4.91)	3.32 (1.96, 5.60)	1.45 (0.85, 2.48)	2.59 (1.25, 5.35)	1.26 (0.33, 4.76)	1.77 (1.18, 2.65)
Second trimester miscarriage	2.92 (1.35, 6.32)	2.01 (1.03, 3.93)	1.90 (0.54, 6.75)	4.33 (2.52, 7.43)	3.38 (1.94, 5.88)	2.10 (0.95, 4.61)	2.71 (1.40, 5.23)	1.72 (0.60, 4.90)	2.28 (1.45, 3.60)
Ectopic pregnancy	1.28 (0.81, 2.02)	0.96 (0.36, 2.57)	-	1.69 (0.65, 4.40)	1.69 (0.65, 4.40)	1.61 (0.79, 3.29)	1.01 (0.22, 4.69)	3.75 (0.66, 21.39)	1.60 (0.92, 2.78)
Preterm delivery	2.98 (2.43, 3.65)	1.62 (0.86, 3.04)	3.15 (1.34, 7.40)	2.93 (2.01, 4.28)	3.11 (2.24, 4.32)	2.83 (1.92, 4.19)	3.69 (2.60, 5.22)	4.93 (3.60, 6.75)	3.50 (2.74, 4.46)
PPROM	3.50 (2.22, 5.54)	-	-	4.66 (1.79, 12.15)	4.66 (1.79, 12.15)	5.42 (1.80, 16.30)	3.77 (1.56, 9.08)	5.80 (1.89, 17.77)	4.66 (2.83, 7.69)
Malpresentation	9.10 (5.88, 14.08)	3.27 (1.66, 6.44)	11.42 (3.74, 34.86)	11.49 (5.24, 25.17)	11.39 (6.24, 20.78)	8.09 (3.14, 20.84)	10.87 (6.68, 17.68)	7.20 (3.09, 16.74)	8.68 (5.82, 12.95)
Cesarean section	2.87 (1.56, 5.26)	2.22 (1.07, 4.61)	5.91 (1.59, 21.95)	4.84 (2.33, 10.02)	5.02 (2.77, 9.10)	1.24 (0.76, 2.03)	5.23 (2.11, 12.96)	7.55 (2.40, 23.72)	3.91 (2.14, 7.13)
Preeclampsia	1.25 (1.07, 1.46)	0.76 (0.18, 3.22)	-	0.25 (0.03, 1.82)	0.25 (0.03, 1.82)	1.27 (0.29, 5.45)	2.83 (0.29, 27.84)	0.45 (0.06, 3.31)	1.49 (0.52, 4.30)
Placental abruption	3.12 (1.58, 6.18)	4.56 (1.03, 20.06)	17.45 (5.05, 60.22) *	5.33 (1.50, 18.95)	9.22 (3.42, 24.82)	7.78 (1.99, 30.45)	6.53 (1.96, 21.78)	2.68 (0.51, 14.17)	6.53 (3.39, 12.61)
Placenta previa	1.56 (0.60, 4.07)	1.68 (0.22, 13.06)	-	1.21 (0.16, 9.33) *	1.21 (0.16, 9.33)	2.71 (0.35, 21.28)	3.59 (1.89, 6.82)	2.18 (0.28, 17.07)	3.37 (1.88, 6.06)
Postpartum hemorrhage	1.02 (0.67, 1.55)	-	-	-	-	-	-	-	-
Recurrent pregnancy loss	2.61 (2.31, 2.94)	-				1.65 (0.09, 32.05) *	2.69 (2.05, 3.52)	0.37 (0.02, 6.25) *	2.63 (2.01, 3.44)
Cervical imcompetence	7.94 (3.81, 16.55)	-	-	9.13 (1.97, 42.33) *	9.13 (1.97, 42.33)	13.69 (2.10, 89.18) *	10.04 (6.30, 15.99)	5.49 (1.63, 18.44)	9.04 (6.11, 13.37)
IUFD	2.06 (1.36, 3.11)	-	-	1.04 (0.12, 9.15)	1.04 (0.12, 9.15)	2.40 (1.27, 4.53)	3.09 (0.38, 24.79)	3.30 (0.38, 28.95)	2.50 (1.39, 4.50)
IUGR or SGA	2.53 (1.77, 3.62)	3.77 (0.92, 15.46)	2.40 (1.13, 5.09)	1.90 (0.89, 4.08)	2.14 (1.26, 3.65)	3.50 (1.24, 9.91)	2.84 (1.68, 4.80)	4.03 (2.00, 8.12)	3.30 (2.29, 4.75)
Perinatal mortality	2.17 (1.46, 3.23)	2.11 (0.79, 5.63)	2.51 (0.82, 7.69)	2.57 (1.08, 6.08)	2.55 (1.29, 5.04)	3.85 (1.61, 9.20)	3.17 (2.08, 4.84)	1.75 (0.77, 3.96)	2.93 (2.15, 4.00)
Low birth weight	1.59 (0.94, 2.68)	1.50 (0.70, 3.25)	-	1.73 (0.91, 3.29)	1.73 (0.91, 3.29)	1.84 (1.08, 3.14)	1.91 (1.12, 3.27)	2.87 (1.38, 5.97)	1.99 (1.38, 2.87)

Values are given as odds ratio [95% confidence interval]. * Only one study presented data on the outcome. CUA, congenital uterine anomalies; PPROM, preterm premature rupture of membranes; IUFD, intrauterine fetal death; IUGR, intrauterine growth restriction; SGA, small for gestational age.

## Data Availability

The data presented in this study are available on request from the corresponding author.

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
