# Peer review of "Reproductive, Obstetric and Neonatal Outcomes in Women with Congenital Uterine Anomalies: A Systematic Review and Meta-Analysis"

_jcm, 2021, doi:10.3390/jcm10214797_

Round 1

Reviewer 1 Report

I read with great interest the manuscript, which falls within the aim of this Journal.  In this study the authors discuss about the impact of congenital uterine anomalies on fertility and obstetrical outcomes which is still a hot topic because the mechanism associated with infertility is still unknown. In my honest opinion, the topic is interesting enough to attract the readers' attention and meets all criteria for publication. The paper is well written and has important clinical messages. Nevertheless, the authors should improve the discussion citing relevant and novel key articles about the surgical treatment of uterine septum. Could the authors give some details on the surgical technique for uterine septum removal?

Authors should consider the following recommendations:

In the last years second- generation techniques have been used in hysteroscopy with good results. These new technological advances in the treatment of septum could reduce the formation of adhesions and consequently reduce the occurrence of septum persistence. Also these new techniques have demonstrated sufficient efficacy in terms of reproductive outcomes.

Please refer to these articles:

1. PMID: 26954490

2. https://doi.org/10.1186/s10397-021-01093-8

Author Response

Reviewer: 1

I read with great interest the manuscript, which falls within the aim of this Journal.  In this study the authors discuss about the impact of congenital uterine anomalies on fertility and obstetrical outcomes which is still a hot topic because the mechanism associated with infertility is still unknown. In my honest opinion, the topic is interesting enough to attract the readers' attention and meets all criteria for publication. The paper is well written and has important clinical messages. Nevertheless, the authors should improve the discussion citing relevant and novel key articles about the surgical treatment of uterine septum. Could the authors give some details on the surgical technique for uterine septum removal?

Authors should consider the following recommendations:

In the last years second- generation techniques have been used in hysteroscopy with good results. These new technological advances in the treatment of septum could reduce the formation of adhesions and consequently reduce the occurrence of septum persistence. Also these new techniques have demonstrated sufficient efficacy in terms of reproductive outcomes.

Please refer to these articles:

  1. PMID: 26954490

  1. https://doi.org/10.1186/s10397-021-01093-8

Reply:

We appreciate the reviewer’s valuable comments. Some researchers have been reported on comparative analysis of reproductive outcomes before and after surgical intervention, but high-quality evidence on the efficacy and safety of surgical treatment to improve reproductive outcomes is lacking. In some cases, hysteroscopic resection of a uterine septum can be offered on an individualized basis for women with septate uterus by experienced specialists.

As the reviewer mentioned, the development of new technology (ex. diode laser) associated with the great improvements in hysteroscopy has led to new interesting therapeutic applications for the resection of the uterine septum. So, we have added the details regarding this comment in the section of “Discussion” as follows:

Various hysteroscopic instruments, including microscissors, forceps, operating loop, electrosurgical needle, and laser energy, have been proposed for surgical correction [73]. Gradually, several methods are being introduced to help the proliferation of endometrial tissue and reduce adhesion formation and minimize intraoperative or postoperative complications [74]. Tissue vaporization and coagulation with low-power diode laser energy can rapidly remove the uterine septum without damaging the underlying myometrium [75,76]. Still, there is a lack of large-scale studies on which technique is better.

Thank you for all your valuable comments. We think the editor and reviewers’ comments have significantly improved the quality of our manuscript by guiding our insights appropriately.

Reviewer 2 Report

Nice work. It shows the extensive research you have done to review all the literature for the available data, although by using all data the heterogenicity increases, which makes the conclusion less powerful. 

Author Response

Thank you for your attention to our manuscript. I look forward to your favorable reply.

Reviewer 3 Report

The authors have conducted a systematic review and metaanalyses regarding risk of dismal reproductive outcome with uterine anomalises.

The methods are described properly, the statistics seem correct and their conclusions based on the results.

The language (English) is satisfactory.

I have only one question: The sentence “Significance levels were set at P<0,05 if the 95% CI did not include 1.” This seems not meaningful. If the CI does not include 1 this is implying a not-significant difference (usually set at P<0,05). Were p-values/statistical tests only performed if the CI included 1? Actually NO p-values have been cited in the manuscript, please clarify this.

Author Response

Reviewer: 3

The authors have conducted a systematic review and metaanalyses regarding risk of dismal reproductive outcome with uterine anomalises.

The methods are described properly, the statistics seem correct and their conclusions based on the results.

The language (English) is satisfactory.

I have only one question: The sentence “Significance levels were set at P<0,05 if the 95% CI did not include 1.” This seems not meaningful. If the CI does not include 1 this is implying a not-significant difference (usually set at P<0,05). Were p-values/statistical tests only performed if the CI included 1? Actually NO p-values have been cited in the manuscript, please clarify this.

Reply:

Thank you for your valuable comment. As the reviewer pointed out, there was a possibility of confusion. Regardless of the CI value, the p-value value was obtained by performing a statistical test in all cases. We have corrected the details regarding this comment in the “Materials and Methods” section as follows and added the p-value to all results:

Statistical significance was set at P <0.05. We performed all statistical analyses using Review Manager (RevMan) 5.4 (The Nordic Cochrane Centre: Copenhagen).

Thank you for all your valuable comments. We think the editor and reviewers’ comments have significantly improved the quality of our manuscript by guiding our insights appropriately.